# 4-Thiazolidinone-Bearing Hybrid Molecules in Anticancer Drug Design

**DOI:** 10.3390/ijms232113135

**Published:** 2022-10-28

**Authors:** Piotr Roszczenko, Serhii Holota, Olga Klaudia Szewczyk, Rostyslav Dudchak, Krzysztof Bielawski, Anna Bielawska, Roman Lesyk

**Affiliations:** 1Department of Biotechnology, Medical University of Bialystok, Kilinskiego 1, 15-089 Bialystok, Poland; 2Department of Pharmaceutical, Organic and Bioorganic Chemistry, Danylo Halytsky Lviv National Medical University, Pekarska 69, 79010 Lviv, Ukraine; 3Department of Synthesis and Technology of Drugs, Medical University of Bialystok, Kilinskiego 1, 15-089 Bialystok, Poland

**Keywords:** molecular hybridization, hybrid molecules, 4-thiazolidinone, approved drugs, natural compounds, privileged heterocycles, anticancer activity

## Abstract

Oncological diseases have currently reached an epidemic scale, especially in industrialized countries. Such a situation has prompted complex studies in medicinal chemistry focused on the research and development of novel effective anticancer drugs. In this review, the data concerning new 4-thiazolidinone-bearing hybrid molecules with potential anticancer activity reported during the period from the years 2017–2022 are summarized. The main emphasis is on the application of molecular hybridization methodologies and strategies in the design of small molecules as anticancer agents. Based on the analyzed data, it was observed that the main directions in this field are the hybridization of scaffolds, the hybrid-pharmacophore approach, and the analogue-based drug design of 4-thiazolidinone cores with early approved drugs, natural compounds, and privileged heterocyclic scaffolds. The mentioned design approaches are effective tools/sources for the generation of hit/lead compounds with anticancer activity and will be relevant to future studies.

## 1. Introduction

Cancer diseases are a colossal problem for humanity, and according to data from the World Health Organization (WHO), they are the major cause of death among people under 70 years [1]. This poses a significant challenges to research focused on the development of novel and more effective pharmaceuticals without the disadvantages/adverse effects of classic/conventional anticancer chemotherapeutics, which usually include a low therapeutic index, development of resistance to treatment, and low bioavailability [2]. The modern medicinal chemistry of anticancer agents actively uses a wide set of tools for updating anticancer agent libraries and pipelines with the aim to achieve the desired biological activity and overcome the limitations of known chemotherapeutics. 

Molecular-hybridization approaches are important synthetic strategies in the design of drug-like small molecules with more powerful and biologically more extensive anticancer properties than the initial compounds [3].

4-Thiazolidinone scaffolds belong to privileged structures in drug design [4]. The most successful molecules from this class of heterocycles include epalrestat [5] and glitazones [6] (Figure 1), which has made a significant contribution to diabetes therapy. Over the last decade, 4-thiazolidinone derivatives have continued to be the focus of research by medicinal chemists and have once again returned to the world pharmaceutical market. Thus, the 5-ylidene derivative of 2-(alkyl)imino-4-thiazolidinone Ponesimod (Ponvory) (Figure 1) was approved by the FDA in 2021 as a potential drug for the treatment of multiple sclerosis and psoriasis [7,8].

4-Thiazolidinone scaffolds are widely used for the development of anticancer agents, and a number of recent reviews have been published dedicated to this area of drug design [9,10]. The works in question include material devoted specifically to derivatives of 2-thioxothiazolidin-4-one (rhodanine) [9] and thiazolidine-2,4-dione [10], which possess antitumor properties. Therefore, this review’s purpose was to highlight the anticancer properties of 2-imino-4-thiazolidinone derivatives and 2-3-disubstituted 4-thiazolidinones as well as several recent works devoted to 2-thioxothiazolidin-4-one (rhodanine) or thiazolidin-2,4-dione derivatives. For the writing of this review, the research articles, short communications, letters, and reports from the scientific databases Scopus (Elsevier), SciFinder (Chemical Abstracts), and PubMed were analyzed from the years 2017 to 2022. Information from patents was not included in the present study. Moreover, the main emphasis of the present review was on the application of molecular hybridization strategies such as the hybridization of scaffolds, the hybrid-pharmacophore approach, analogue-based drug design in synthesis, and the development of novel potent antitumor agents with 4-thiazolidinone scaffolds.

## 2. Approved Drug–4-Thiazolidinones Hybrids

A series of publications reported the successful application of molecular hybridization methods and strategies for the design of novel anticancer agents based on obtaining hybrids of the type “approved drugs-4-thiazolidinone scaffolds”. The drugs from different pharmacological groups were used as parent molecules for further hybridization and development.

### 2.1. Anticancer Drug–4-Thiazolidinones Hybrids

Türe et al. [11] reported on the design and synthesis of a series of novel hybrid molecules containing imatinib (Gleevec) moiety. Imatinib (Figure 2) was the first-in-class protein kinase inhibitor approved by the FDA in 2001 and had a revolutionary impact on the treatment of most cases of chronic myeloid leukemia (CML) [12,13]. Obtained by Türe et al., 5-benzylidene-2-arylimino-4-thiazolidinone-bearing imatinib analogs were screened for their antimitotic activity on K562 (chronic myeloid leukemia), PC3 (prostate cancer), and SHSY-5Y (neuroblastoma) cell lines, and the three most potent cytotoxic hybrids, **1–3**, were identified. Compounds **1–3** induced apoptotic cell death. Moreover, cycle arrest in the G0/G1 phase was caused by compounds **2** and **3**, while compound **1** inhibited the cell cycle in the G2/M phase. Hybrid molecules **1** and **3** proved to have stronger genotoxic effects than imatinib against K562 cells [11].

### 2.2. NSAID–4-Thiazolidinones Hybrids

Recent decades have been marked by an increased interest in the use of potential NSAIDs for the treatment of cancer or using NSAIDs scaffolds/molecules for the design of new antimitotic agents [14]. 

Ramadan W.S. et al. [15] used a hybridization approach involving anti-inflammatory drug-5-aminosalicylic acid and 5-ylidene-4-thiazolidinone scaffolds with the aim of designing new promising anti-cancer agents (Figure 3). Synthesized hybrids were tested on a panel of seven cancer cell lines, and two molecules, **4** and **5**, were found to be efficient in some types of cancer compared to the effect of doxorubicin (reference drug), with the highest activity level detected on the MCF7 line, with IC_50_ values of 0.31 and 0.30 µM, respectively. Furthermore, tumor specificity and minimal effects on normal fibroblasts were observed for **4** and **5**. In vitro studies of the molecular mechanisms of action for **4** and **5** revealed the induction of DNA damage, cell cycle arrest in the G2/M phase, and the induction of apoptosis as indicated by annexin-V staining and the activation of caspases. Moreover, it was found that both compounds modulate the expression and activity of several factors in the DNA damage response pathway, cyclins/cyclin-dependent kinases, and CDC25 phosphatase.

NSAID–diclofenac was used for the design of potential anticancer agents by Shepeta et al. [16]. A series of diclofenac-4-thiazolidinones hybrids with a hydrazine linker in the molecules were synthesized and tested on antitumor activity in a 60 line screening program (DTP NCI) [17,18,19,20]. Two novel hybrids, **6** and **7** (Figure 3), displayed slight anticancer activity, and an in vitro cytostatic effect was observed on the leukemia CCRF-CEM, with a percentage of growth (GP%) of 56.82% for molecule **6**, on the non-small-cell lung cancer NCIH522, with a GP of 54.32%, and on colon cancer HCT-116, with a GP% of 57.71% of cell lines for hybrid **7**.

Holota et al. [21] reported on pyrazolin-5-one bearing 4-thiazolidinone hybrid **8** (Figure 3) with moderate anticancer activity, with GP% values of 68 % and 87 % on the NCI-H460 (Lung cancer) and SF-268 (CNS cancer) lines, respectively, at 10 μM concentration.

### 2.3. Antibacterial Drug–4-Thiazolidinones Hybrids

A series of novel 4-thiazolidinone-sulfanilamide hybrids incorporating substituted indolin-2-one moieties were designed as possible human carbonic anhydrase (hCA) inhibitors by Eldehna et al. [22]. The new hybrids were evaluated in vitro for their inhibitory activity against hCA I, II, IV, and IX, and all molecules were active in variable degrees. Moreover, all compounds were examined for their anti-proliferative activity against breast cancer MCF-7 and colorectal cancer Caco-2 cell lines, and hybrid **9** (Figure 4) was found to be the most potent against MCF-7, with an IC_50_ of 3.96 ± 0.21 µM. In-depth studies showed that hybrid **9** provoked the intrinsic apoptotic mitochondrial pathway in MCF-7 cells, which was evidenced by the enhanced expression of the pro-apoptotic protein Bax and the reduced expression of the anti-apoptotic protein Bcl-2 as well as up-regulated active caspase-3 and caspase-9, cytochrome C, and p53 levels.

Multifunctional sulfanilamide-bearing hybrids were developed and tested against HepG2 and MDA-MB-231 cancer cell lines by Sunil Kumar et al. [23]. Among the tested hybrids, compound **10** (Figure 4) was identified as potent against the MBA-MB-231 cell line and showed the highest cytotoxic activity, with an IC_50_ value of 17.45 µM, which was higher than in the case of the standard drug cisplatin.

Special attention from the point of view of analog- and fragment-based anticancer drug design is warranted in the case of the molecule ciminalum, which is a *p*-nitro-α-chlorocinnamic aldehyde or (2*Z*)-2-chloro-3-(4-nitrophenyl)prop-2-enal (CAS 3626-97-9) and was used as a drug in medical practice in the former Soviet Union (Figure 5) as an active antimicrobial agent against Gram-positive and Gram-negative microorganisms.

The diversity-oriented approach was applied for the synthesis of ciminalum-4-thiazolidinone hybrid molecules, and a series of derivatives were reported by Buzun and Finiuk [24,25,26]. For such a hybrid type, SAR analysis revealed that the presence of carboxylic acids [25], p-hydroxyphenyl [24], or (4-hydroxyphenyl)-pyrrolidine-2,5-dione [26] substituents at position 3 of the 4-thiazolidinone ring determines the most effective anti-tumor activity, while the absence of a substituent or presence of an additional ciminalum substituent at position 3 leads to a weakening in activity. Among the synthesized and screened hybrids, derivative **11** (Figure 6) was cytotoxic toward MCF-7 and MDA-MB-231 breast cancer cell lines, with IC_50_ values of 5.02 µM and 15.24 µM, respectively [25]. The hybrid **11** induced the extrinsic and intrinsic apoptotic pathways and caused a reduction in topoisomerase II concentration in both tested human breast cancer cell lines. The affinity of hybrid **11** to topoisomerase II was also confirmed by docking simulations. In addition, molecule **11** caused a decrease in the levels of Beklin-1, LC3A (the microtubule-associated protein 1A/1B light chain 3A), and LC3B (the microtubule-associated protein 1A/1B light chain 3B), suggesting its influence on the autophagy process.

A ciminalum–4-thiazolidinone hybrid **12** [24] (Figure 6) possessed a high activity level in NCI 60-Cell-line screening [17,18,19,20] (MG_MID GI_50_ = 1.57 µM and TGI = 13.3 µM). The analysis of activity inside cancer panels revealed a certain sensitivity profile in the GI_50_ concentration range of < 0.01–0.02 µM toward leukemia (MOLT-4, SR), colon cancer (SW-620), CNS cancer (SF-539), and melanoma (SK-MEL-5) cell lines. In the same paper [24], the high cytotoxicity of 3-((Z)-5-((Z)-2-chloro-3-(4-nitrophenyl)allylidene)-4-oxo-2-thioxothiazolidin-3-yl)propanoic acid **13** (Figure 6) against cell lines of gastric cancer (AGS), human colon cancer (DLD-1), and breast cancers (MCF-7 and MDA-MB-231), with values of GI_50_ 2.69, 3.67, 3.62, and 1.63 µM, respectively, was established.

Finiuk et al. [26] reported on ciminalum–4-thiazolidinone hybrids containing phenyl-pyrrolidine-2,5-dione moieties in the molecules. The hit-compounds **14** and **15** (Figure 6), possessing micromolar cytotoxic activity towards leukemia, colon cancer, CNS, and ovarian cancer cell lines, were identified following a 60 lines NCI protocol with total MG_MID GI_50_ values of 1.76 and 1.73 μM, respectively. Treatment by **14** and **15** led to the significant inhibition of T-leukemia cells of the Jurkat line, with high selectivity indices of 29.4 and 56.6, respectively. Hybrids **14** and **15** altered the levels of mitochondrial apoptosis-associated proteins (Bax and EndoG, Bcl-2) and caused apoptosis toward human T-leukemia cells of the Jurkat line. However, the application of **14** and **15** did not affect the morphology and the DNA intactness in mitogen-activated lymphocytes of the peripheral blood of healthy human donors.

## 3. Natural Compound Scaffold–4-Thiazolidinones Hybrids

The structural motifs of natural products always serve as an excellent source for drug-like molecules from the perspective of the design of anticancer agents [27,28].

### 3.1. Monoterpene–4-Thiazolidinone Hybrids

Fawzi et al. [29] synthesized 4-acetyl-1-methylcyclohexenothiosemicarbazones and 4-acetyl-1-methylcyclohexeno-2-imino-4-thiazolidinone hybrids based on a monoterpene backbone structure, that of limona ketone, containing hydrazone linker in the molecules (Figure 7). The cytotoxicity of the obtained compounds was evaluated on HT-1080, A549, and MCF-7 cell lines. Compounds **16** and **17** (Figure 7) proved to be the most cytotoxic on HT-1080 lines, demonstrating IC_50_s of 15.85 µM and 16.13 µM, respectively. Through molecular docking, these compounds were found to form stable ligand-caspase-3 complexes. In addition, compound **17** was revealed to be the most potent in generating apoptosis and caspases-3/7 activation, including S-phase cell cycle inhibition in HT-1080 cells, while compound **16** exhibited a lower degree of apoptosis induction compared to compound **17**, in turn inducing G0/G1 phase arrest in the same cells.

Oubella et al. [30] developed hybrids containing the monoterpene-carvone skeleton and 4-thiazolidinone scaffolds linked by the 1,2,3-triazole ring (Figure 8). The novel hybrids were screened for their anticancer activity against four cell lines: breast MCF-7 and MDA-MB-231, lung A-549 and fibrosarcoma HT-1080. The highest cytotoxic activity was demonstrated by compounds **18** and **19** (Figure 8). Expanding the study, the researchers proved that such activity is generated through the induction of apoptosis via caspase-3, as confirmed by molecular docking, and effects on the cell cycle, specifically arresting cells in S and G2/M phases in HT-1080 and A-549 cells, respectively.

Applying thymol as part of the structure of novel hybrids with 4-thiazolidinone scaffolds is a promising approach. A synthesis and anticancer activity evaluation of a series of thymol–4-thiazolidinone hybrids was reported by El-Miligy et al. [31]. Among the new hybrids compounds, **20–22** (Figure 9) exhibited the highest activity against human colorectal cancer (CRC) cell lines (Caco-2 and HCT-116) at doses less than their EC_100_ on normal human cells. Furthermore, compounds **21** and **22** induced apoptosis-dependent death and caspase activation by >50% in human CRC cell lines. Furthermore, compounds **20–22** showed in vitro inhibitory activity against both PIM-1/2 kinases comparable to the reference, staurosporine.

### 3.2. Coumarin 4-Thiazolidinones Hybrids

The application of the coumarin motif is an attractive direction in anticancer drug design [32]. Following this strategy, Sigalapalli et al. [33] reported the synthesis of novel molecular hybrids of 4-thiazolidinone–umbelliferone (7-hydroxycoumarin) as prominent cytotoxic agents. The most active derivative, **23** (Figure 10), showed the highest potency against A549 cells, with an IC_50_ value of 0.96 ± 1.09 µM and a selectivity index of 51.7. The in-depth study of hybrid **23** showed apoptosis induction by the annexin-v/PI dual staining assay; its effect on different phases of the cell cycle; the inhibition of tubulin polymerization at an IC_50_ value of 2.65 ± 0.47 µM; and its effective binding with CT-DNA. In silico experiments revealed a prominent binding affinity towards the α/β-tubulin receptor with remarkable protein–ligand interactions and binding energy in the case of **23**.

Thacker et al. [34] reported the synthesis of coumarin–4-thiazolidinone hybrids with pyrazole linker in the molecules as potential hCA inhibitors. The results obtained by the authors indicated the selective inhibition by synthesized hybrids of the tumor-associated isoforms hCA IX and XII but not in the case of the off-target isoforms, hCA I and II. Compound **24** (Figure 10), with the best hCA XII inhibition constant value, 61.5 nM, was considered the lead compound for further developing selective and potent hCA IX inhibitors.

The synthesis and evaluation of antiproliferative activity in A549 (lung cancer), MDA-MB-231 and BT-474 (breast cancer), HepG2 (liver cancer), and HCT-116 (colon cancer) cell lines using an MTT of coumarin-2–iminothiazolidin-4-one hybrids were reported by Sigalapalli et al. [35]. Hybrid molecule **25** (Figure 10) showed excellent anti-proliferative activity on the breast cancer cell lines MDA-MB-231 and BT-474 (IC_50_ of 0.95 ± 1.88 and 1.22 ± 0.08 µM, respectively) and lung cancer cell line A549 (IC_50_ = 1.28 ± 0.98 μM). The cell cycle analysis disclosed that **25** showed significant G2/M phase arrest in MDA-MB-231 cells. Furthermore, hybrid **25** significantly inhibited tubulin polymerization, with an IC_50_ value of 3.54 ± 0.2 µM, and caused apoptosis-mediated cell death in MDA-MB-231 cells. In silico studies inferred that hybrid **25** binds at the colchicine binding site of the tubulin, with prominent binding affinity.

### 3.3. Hybrids of 4-Thiazolidinones with Steroidal Skeleton

Steroids are important and physiologically significant natural compounds, opening up the possibility of structural tuning with the aim of achieving and developing valuable pharmacological properties including potent anticancer activity [36]. 

Živković et al. [37] designed a series of mono- and bis-4-thiazolidinone-containing hybrids with androstene derivatives. For the synthesized hybrids, anticancer activity studies were performed on six cancer lines: HeLa (cervical adenocarcinoma), K562 (chronic myelogenous leukemia), MDA-MB-453 (breast carcinoma), MDA-MB-361 (breast adenocarcinoma), LS174 (colon adenocarcinoma), A549 (lung carcinoma), and MRC-5 (normal lung fibroblast line).

All the obtained molecules exhibited selective concentration-dependent cytotoxicity on all tested lines. The strongest response to the tested compound was observed on the K562 and HeLa cell lines, where the IC_50_ of the majority oscillated around 10 µM, with the IC_50_ of the cisplatin reference compound for these lines being 5.7 µM and 5.2 µM, respectively. Compared to the positive control, these compounds exhibited lower toxicity to normal cells. Hybrids **26** and **27** were found to be the most active (Figure 11) and inhibited the cell cycle in HeLa cells in subG1, S, and G2/M phases and induced apoptosis by an intrinsic and extrinsic pathway.

## 4. Privileged Heterocyclic Scaffolds–4-Thiazolidinones Hybrid Molecules

Hybrids with 4-thiazolidinone scaffolds as potent anticancer agents are the most intensively studied type of hybrids in recent years and represents the largest group of molecules among recently published data.

### 4.1. Furan–4-Thiazolidinone Hybrids

Tahmasvand et al. [38] reported on the design and synthesis of eight novel 4-thiazolidinone hybrids which were evaluated for their in vitro anticancer activity against MDA-MB-231, HT-29, and HepG2 cell lines. The furan-bearing hybrid **28** (Figure 12) demonstrated very strong activity against the three cell lines after 72 h with IC_50_ values of 1.9 µM, 6.5 µM, and 5.4 µM, respectively, which was comparable to the control sample with the reference compound doxorubicin.

Hybrid **28** induced apoptosis through the regulation of pro-caspase 3 and cell cycle arrest in the G1/S phase. In addition, compound **28** downregulated MMP-9 mRNA expression in MDA-MB-231 cells. In vivo studies performed for hybrid **28** on mouse line 4T1 demonstrated a dose-dependent reduction in mammary tumor growth without weight loss in the test model.

### 4.2. Pyrrole–4-Thiazolidinone Hybrids

A pyrrole core was used as a pharmacophore in the design of potential antimitotic agents by Shawky et al. [39]. The synthesized hybrid molecules **29–31** (Figure 13) exhibited potent cytotoxicity, with an IC_50_ of 0.10–0.60 µM against three cancer cell lines: MCF-7, A2780, and HT29. Additional studies showed that hybrid **29** induced G1 cell cycle arrest and apoptosis in MCF-7 cells and displayed inhibitory activity against CDK2, with an IC_50_ of 0.63 µM.

The pyrrolidine-2,5-dione–4-thiazolidinone hybrid **32** (Figure 13) was reported by Finiuk et al. [26]. Compound **32** exhibited moderate activity in the 60 lines screening DTP NCI [17,18,19,20], with a GP% value of 68.20 % on the MDA-MB-445 (Melanoma) cell line and with a GI_50_ value of 47.50 µM towards the HeLa cell line. Additionally, hybrid **32** demonstrated low toxicity towards normal human keratinocytes of the HaCaT line, with a GI_50_ value of >100 μM.

### 4.3. Pyrazole–4-Thiazolidinone Hybrids

A series of 4-thiazolidinone-pyrazole hybrids connected via a ylidene linker were synthesized by Bhat et al. [40]. An in vitro screening of the synthesized hybrids revealed that compounds **33** and **34** (Figure 14) exhibited potent anticancer properties against the MDA-MB-231 cell line, with IC_50_ values 24.6 and 29.8 μM, respectively (MTT assay). Furthermore, molecule **34** exhibited superior protection of normal HDF cells than **33**. Additionally, the anticancer potency of the compounds was justified by using in silico studies. SAR analysis concluded by the authors suggests the importance of the presence of dichlorosubstituted arylamino moiety in position two of the 4-thiazolidinone core in enhancing potency.

Mushtaque et al. [41] reported on the synthesis and anticancer assessment of a pyrazole-bearing hybrid molecule, **35** (Figure 14). Docking studies for the hybrid **35** revealed that the molecule could interact through the minor groove of DNA. Additionally, MTT-assay data showed that compound **35** was non-toxic up to concentrations of 282.32 µg/mL against the cancerous MCF-7 cell line and 200 µg/mL against Siha cells.

### 4.4. Pyrazole-Purine–4-Thiazolidinone Hybrids

Hybrids containing pyrazole, purine, and 4-thiazolidinone scaffolds were developed by Afifi et al. [42] as potential anticancer agents.

The most potent cytotoxic activity among all the analyzed hybrids was shown by compound **36** (Figure 15), with IC_50_ values of 18.85 µM, 23.43 µM, 23.08 µM, 23.08 µM, and 18.50 µM for the A549, MCF-7, HepG-2, Caco-2, and PC-3 cell lines, respectively, whereas the values for the reference compound 5-fluorouracil oscillated around 90 µM. As with cytotoxic activity, compound **36** showed the strongest antioxidant properties; however, these were slightly weaker than in the case of the reference compound, ascorbic acid.

### 4.5. Pyrazole-Piperazine–4-Thiazolidinone Hybrids

In study [43], new pyrazole-based 4-thiazolidinones hybrids linked with piperazine moiety were reported as promising selective VEGFR2 tyrosine kinase inhibitors. Among all the tested compounds, the derivatives **37–39** (Figure 15) were found to be the most active and selective against the HepG-2 cancer cell line, with IC_50_ values 0.06 ± 0.003 µM, 0.03 ± 0.006 µM, and 0.06 ± 0.004 µM, respectively, and with the significant selectivity indices: 8.09, 11.40, and 4.37, respectively. Also, the synthesized hybrids showed potent VEGFR2 tyrosine kinase inhibitory activities, with lower IC_50_ values than that of the reference drug, staurosporine, with this potency being confirmed by additional docking studies which demonstrated that **37–39** could act as inhibitors of VEGFR2 tyrosine kinase via the stabilization of the enzyme inactive DFG-out conformation.

### 4.6. Pyrazoline-Pyrrole–4-Thiazolidinone Hybrids

Safaa I. Elewa et al. [44] reported the synthesis and in vitro and in silico study of novel pyrazoline–4-thiazolidinone hybrids with a pyrrole ring in the molecules. The newly synthesized derivatives **40** and **41** (Figure 16) exhibited remarkable cytotoxic activities with promising IC_50_ values in terms of cytotoxicity against MCF-7 and HCT-116 cells using the MTT assay.

Hybrid **41** exhibited potent cytotoxicity against MCF-7 and HCT-116 cell lines, with IC_50_ values of 5.05 and 3.08 µM, compared to respective doxorubicin IC_50_ values of 7.27 and 8.92 µM. Compound **40** showed a lower activity level, with IC_50_ values of 15.67 on MCF-7 and 19.39 µM on HCT-116 cell lines. Both hybrids **40** and **41** were found to be nontoxic against normal WISH cells, with IC_50_ values of more than 50.0 µM. Additionally, all the molecules reported in the article were screened for their binding activity toward CDK-2 kinase activity using a molecular docking study, and the obtained results suggest the tested compounds could be possible CDK-2 inhibitors. 

### 4.7. Isoxazole–4-Thiazolidinone Hybrids

The synthesis of the hybrid 2-(4-fluorophenyl)-3-(5-methylisoxazol-3-yl)thiazolidin-4-one, **42** (Figure 17), as a potential antimitotic agent was reported by Ramesh et al. [45].

The results of an in vitro cytotoxicity assessment employing the MTT assay as per the ATCC protocol, using three human cancer cell lines, HeLa (cervical cancer), MCF-7 (breast cancer), A549 (human lung cancer), and HEK-293 (normal human embryonic kidney cells), indicated that hybrid **42** exhibited excellent anticancer activity against MCF7 and displayed less cytotoxicity towards normal HEK293 cell lines. Hybrid molecule **42** inhibited the cell proliferation of HeLa by 58%, 87%, and 94% at 6.25 µM, 25 µM, and 50 µM concentrations, respectively. In terms of inhibition, 59% and 65% inhibition were observed at a 50 µM concentration on breast cancer (MCF7) and lung cancer (A549) cell lines, respectively, by compound **42**. At lower concentrations, compound **42** has less effect on normal cell lines (HEK293). Additionally, a molecular docking study explored the binding mode and possible interactions between the synthesized compound and the ATP binding site of the EGFR kinase domain.

### 4.8. Thiazole–4-Thiazolidinone Hybrids

A series of 5-enamine–4-thiazolidinones with phenylalanine moiety in the molecules was designed and reported by Holota et al. [46]. Among the synthesized derivatives, the hybrid **43** (Figure 18), with a thiazole-bearing scaffold in the molecule, was found to be highly active in the 60 lines screening. Compound **43** demonstrated inhibition activity against (GI_50_ < 10 μM) against all 59 human tumor cell lines, with average GI_50_/TGI/LC_50_ values of 2.57/57.27/94.71 μM, respectively. 

Preliminary SAR provided by the authors revealed that the presence of thiazole substituent is crucial for anticancer activity whereas a change in the hydrogen atom, aniline derivatives, or 1,2,4-triazole nucleus leads to the disappearance of the effect.

### 4.9. Triazole–4-Thiazolidinone Hybrids

Two series of merged 1,2,4-triazole–4-thiazolidinone hybrids with ylidene and enamine linkers in the molecules (Figure 19) were designed and synthesized by Holota et al. [47]. The obtained hybrids were studied for their antitumor activity in an NCI 60 lines screening [17,18,19,20] and a number of the compounds presented excellent anticancer properties at 10 µM. Derivatives **44** and **45** were found to be the most active against cancer cell lines, with total mean GI_50_ values of 3.54 µM (**44**) and 10.96 µM (**45**), without causing toxicity to normal somatic HEK293 cells, with IC_50_ values of 28.99 µM (**44**) and 24.43 µM (**45**).

An SAR analysis performed by authors revealed that both the presence and position of a chlorine atom in the benzylidene area of molecules play a crucial role in the realization of the anticancer effect [47].

### 4.10. Pyridine–4-Thiazolidinone Hybrids

A series of 17 novel pyridine-thiazolidinone hybrids were synthesized, characterized, and evaluated as potential hCA inhibitors by Ansari et al. [48]. Of all the synthesized hybrids, derivatives **46** and **47** (Figure 20) had the most potent CAIX inhibition activity in the esterase assay, with IC_50_ values of 1.61 µM and 1.84 µM, respectively. The evaluated cytotoxicity for **46** and **47** on a HEK-293 was 249.6µM and 230.4µM, respectively. The evaluated anticancer activities against MCF-7 (breast cancer cell line) and HepG-2 (liver cancer cell line) were IC_50_ values of 13.0 µM and 19.2 µM, respectively, for hybrid **46** and values of 12.4 µM and 16.2 µM, respectively, for **47**.

In [49], the synthesis and study of molecules **48** and **49** (Figure 20), as modified analogues of hybrids **46** and **47,** was reported. The authors applied the bioisosteric replacement of the pyridine core by the substituted benzene ring for the design of novel target molecules with desirable anticancer properties. Despite the promising evaluated in silico features, the newly synthesized analogues **48** and **49** were found to be less active compared to the hybrids **46** and **47** against the HepG-2 cell line.

The molecular hybridization approach was applied for the synthesis of pyridine–4-thiazolidinone hybrids with aim of the synthesis of potential antiglioblastoma agents by Campos et al. [50]. Among sixteen synthesized compounds, derivatives **50–52** (Figure 20) displayed potent antitumor activities against the tested glioblastoma cell lines and exhibited IC_50_ values 2.17, 6.24, and 2.93 μM, respectively, in the C6 cell line, well below the standard drug temozolomide. The mechanism of action studies demonstrated that derivatives **50** and **52** induced apoptosis, significantly increasing the percentage of cells in the Sub-G1 phase in the absence of necrosis. Consistent with these results, the caspase-3/7 assay revealed that derivative **50** presents pro-apoptotic activity due to the significant stimulation of caspases-3/7. Moreover, hybrids **50–52** increased antioxidant defense, decreased reactive oxygen species (ROS) production, and modulated redox status.

### 4.11. Pyridine-Piperazine–4-Thiazolidinone Hybrids

The attempt to design potential antimitotic agents following the hybridization of pyridine, piperazine, and 4-thiazolidinone scaffolds was reported by Demirci et al. [51]. The synthesized hybrids were evaluated on the PC-3, DU145, and LNCaP prostate cancer cell lines. All the studied molecules possessed satisfactory calculated pharmacokinetic properties. Compound **53** (Figure 21) showed the most potent activity level inside the tested series, with an IC_50_ of 36.75 µM on the PC-3 cancer line, while the other cancer cells tested showed no response to the compounds used (IC_50_ > 500 µM).

### 4.12. Pyridine-Thiazole–4-Thiazolidinone Hybrids

Hussein et al. [52] applied a hybrid-pharmacophore approach for the design of novel biologically active 4-thiazolidinones. Among the synthesized molecules, compound **54** (Figure 21) exhibited the most potent cytotoxic activity and was found to be more effective on the WI-38 normal human lung fibroblast cell line, with an IC_50_ of 92 µg/mL, but not as effective on the A549 and MDA-MB-31 cancer cells, with IC_50_ values of 357 µg/mL and 505 µg/mL, respectively.

### 4.13. Indole–4-Thiazolidinone Hybrids

Indole scaffolds are often used for the design and development of 4-thiazolidinone hybrids as anticancer agents. Sigalapalli et al. [53] reported on the synthesis of a small library of hybrid molecules containing 4-thiazolidinone and indole scaffolds linked by an aliphatic linker with aim of the designing tubulin polymerization inhibitors. The novel compounds were tested on five human cell lines of the lung (A549, NCI-H460), breast (MDA-MB-231), and colon (HCT-29, HCT-15), using podophyllotoxin as a reference compound. The synthesized compound **55** (Figure 22) showed the strongest activity against both colon cancer cell lines, with an IC_50_ value of 0.92µM for HCT-15. Meanwhile, against the normal human lung epithelial cell line (L132), the IC_50_ value for the compound was 10.84 µM, indicating that compound **55** is 10-fold more selective against the colon cell line. SARs analysis performed by the authors revealed that the antiproliferative activity of phenethyl-thiazolidinone-indole hybrids is higher than that of benzyl-thiazolidinone-indole ones. Furthermore, it was observed that hybrids without a substituent at the nitrogen atom into indole core are more active than N-alkyl/aryl-substituted ones. In addition, the presence of the methoxy group at the phenyl ring of the side chain of hybrid compounds leads to higher activity than halogen substituents. Extended analyses on compound **55** revealed that the compound inhibited the cell cycle of HCT-15 cells in sub G1 and G2/M phases, additionally inhibited tubulin more strongly compared to the reference compound, induced intracellular ROS, and decreased the mitochondrial potential, causing cell death by apoptosis [53].

Oliveira et al. [54] reported on the synthesis of novel 4-thiazolidinone derivatives with an indole backbone. Hybrid **56** (Figure 22) possessed the most potent anticancer activity and was active against the MCF-7 cell line, with an IC_50_ of 6.06 µM, the OVCAR-3 cell line, with an IC_50_ of 5.12 µM, and the HaCat cell line, with IC_50_ of 6.23 µM.

A series of novel indole-azolidinone hybrids with 5-fluoro-3-formyl-1*H*-indole-2-carboxylic acid methyl ester scaffolds were synthesized via the Knoevenagel reaction [55]. Among the synthesized compounds, hybrid **57** (Figure 22) was found to be the most active and exhibited toxicity in the 60 lines screening, with total GI_50_ of 0.45/0.65 µM and with IC_50_ values toward cancer cells MCF-7-0.70 µM, HCT116-0.80 µM, A549-9.70 µM, and HepG2-12.00 µM. Meanwhile, the non-malignant cells (human keratinocytes of HaCaT line and murine embryonic fibroblasts of Balb/c 3T3 line) possessed moderate sensitivity to hybrid **57**. Furthermore, compound **57** induced apoptosis in studied tumor cells via caspase-3-, PARP1-, and Bax-dependent mechanisms; however, it did not affect the G1/S transition in HepG2 cells. Additionally, hybrid **57** impaired nuclear DNA in HepG2, HCT116, and MCF-7 cells without intercalating this biomolecule, but much fewer DNA damage events were induced by **57** in normal Balb/c 3T3 fibroblasts compared with HepG2 carcinoma cells.

### 4.14. Isatine–4-Thiazolidinone Hybrids

Isatin (Figure 23) is an endogenously occurring molecule in human tissues, based on the structure of which tyrosine kinase inhibitor Sunitinib (Sutent^®^) [56] was established. Sunitinib was approved by the FDA in 2006 for the treatment of the imatinib-resistant cancers advanced metastatic renal cell carcinoma and gastrointestinal stromal tumors. 

El-Naggar et al. [57] reported a series of isatine–4-thiazolidinone hybrids, among which molecule **58** (Figure 23) possessed strong cytotoxicity, with IC_50_ values of 7.6 µM and 8.4 µM on the MDA-MB-231 and MCF-7 cell lines, respectively. Furthermore, compound **58** was active on WI-38 (nontumorigenic human lung fibroblast cell line) and MCF-10A (human breast epithelial cell line), with IC_50_ values of 49.1 µM and 73.1 µM, respectively. In addition, hybrid **58** significantly induced the expression of the pro-apoptotic protein Bax, inhibited the expression of the anti-apoptotic protein Bcl-2, and increased the level of caspase-3 compared to the control sample.

Fouad et al. [58] developed a series of isatine–4-thiazolidinone conjugates as potential antimitotic agents. Compound **59** (Figure 23) was effective against colorectal cancer cell line KM12, melanoma cell line UACC-62, ovarian cancer cell line IGROV1, cancer cell line SK-OV-3, and renal cell carcinoma lines (A498, ACHN, CAKI-1, RXF393, and UO-31). Compounds’ activity towards CAKI-1 and UO-31, with IC_50_ values of 4.74 and 3.99 µM, equal to the cytotoxic potency of the reference compound sunitinib. The derivative **59** arrested the cell cycle in the G2/M and pre-G1 phase and inhibited cyclin-dependent kinase (CDK) activity. In addition, in silico and in vivo studies demonstrated that compound **59** has an excellent physicochemical and pharmacokinetic profile (which PK and docking can develop).

Szychowski et al. [59] reported studies on the anticancer potential of an isatine–4-thiazolidinone hybrid modified with a 3,5-diaryl-2-pyrazoline scaffold **60** (Figure 23). The impact of hybrid **60** on cytotoxicity, the apoptotic process, and metabolism in the human squamous carcinoma (SCC-15) cell line was evaluated. The results obtained by the authors showed that the studied molecule **60** exhibits both cytotoxic and proapoptotic properties at a 10–100 µM concentration range. The activation of caspase-3 was caused by **60** and was accompanied by an LDH release and resazurin reduction. Additionally, increased DCF fluorescence in a wide range of concentrations after 6, 24, and 48 h of exposure was observed after treatment with **60,** which suggests the stimulation of ROS production. The properties of **60** established by the authors make it a promising tool for further research as an anti-cancer agent.

### 4.15. Quinoline–4-Thiazolidinone Hybrids

Batran et al. [60] reported on the synthesis of quinoline–4-thiazolidinone hybrids as potential antimitotic agents. Molecule **61** (Figure 24) was active against the MCF-7 breast cancer cell line, with an IC_50_ value of 98.79 µM.

Nafie et al. [61] studied another series of quinolone–4-thiazolidinone hybrids. Five of the ten compounds showed strong cytotoxic activity against the HCT-116 cell line, comparable to 5-FU, while compound **62** (Figure 24) was the most potent cytotoxic, excluding effects on normal intestinal FHC cells. The compound’s inhibitory activity against EGFR was also confirmed, where it acted at a similar level to the reference compound erlotinib. In addition, hybrid **62** caused the induction of apoptosis and cell cycle arrest in the G2 and S phases. Moreover, it was confirmed to increase the expression levels of p53, PUMA, Bax, caspase-3, -8, and -9 protein genes, and the downregulation of Bcl-2. In in vivo studies, the LD_50_ value was established, which was 6 mg/kg BW. It was observed that the use of this hybrid causes a reduction in tumor size comparable to 5-FU (the inhibition ratio was 52.92% for **62** and 57.16% for 5-FU), with the preservation of biochemical and histochemical structures being close to normal.

A new series of quinolone–4-thiazolidinone hybrid molecules were reported by Kumar et al. [62]. All the compounds were screened in vitro for their anticancer activities against MDA-MB-231 and MCF-7 cell lines using an MTT assay. Derivative **63** (Figure 24) was found to be the most potent against the MDA-MB-231 cell line, with an IC_50_ 8.16 of μM, and, at the same time, non-toxic to the human normal kidney HEK 293 cell line, with an IC_50_ of 846.93 μM. Additionally, it was established in the in silico studies (docking, molecular dynamic studies) that hybrid **63** arrested the cell cycle at the G2/M phase and possesses binding properties with N-acetyl transferase (hNAT-1) protein. The authors suggest that the identified quinolinone–4-thiazolidinone hybrid **63** could be considered as a new chemotype targeting hNAT-1 and could be used for hit/lead compound generation in anticancer drug design and discovery.

Qi et al. [63,64,65,66,67] reported on the application of the hybrid-pharmacophore approach using a 4-phenoxy-6,7-dimethoxyquinoline moiety as a key part of foretinib and cabozatinib molecules in the design of novel multi-tyrosine kinase inhibitors based on 4-thiazolidinone core (Figure 25). 

Structural modification of phenyl rings A and B as well as quinoline ring C was applied (Figure 25) in the molecular design. As a result, hit-compounds with inhibitory activity toward mesenchymale epithelial transition factor (c-Met) (**64,65,68**) and toward HT-29 (human colon cancer cell line) (**66–68**) were identified.

### 4.16. Quinazoline–4-Thiazolidinone Hybrids

A series of hybrids with a quinazoline scaffold was synthesized by Samridhi Thakral et al. [68] (Figure 26). The synthesized compounds were screened for their in vitro cytotoxic and growth-inhibitory activities against MCF-7 and Hep-G2 cell lines using the MTT assay method in comparison with the activity of the known anticancer drug doxorubicin. The hybrid with a chlorine atom at the *m*-position in the phenyl ring attached to the 4-thiazolidinone nucleus at position two (**69**) was found to be the most active against the hepatic cell line, with an IC_50_ of 1.79 µg/mL. 

The presence of the methoxy-group at the *p*-position in the mentioned phenyl ring (**70**) leads to activity against breast cancer cell line MCF-7, with an IC_50_ of 1.94 µg/mL. The IC_50_ value for the standard drug in this study was found to be 0.09 µg/ml.

### 4.17. Benzimidazole–4-Thiazolidinone Hybrids

Sharma et al. [69] reported design and synthesis as potential apoptotic agents of a series of new benzimidazole–4-thiazolidinedione hybrid molecules. All the new twenty synthesized hybrids were evaluated for their in vitro cytotoxic potential against selected human cancer cell lines: breast (MDA-MB-231), prostate (PC-3), cervical (HeLa), lung (A549), bone (HT1080), and a normal kidney cells (HeK-293T) using the MTT assay. Eleven of the twenty described hybrids were active in the nanomolar concentration range (IC_50_ values from 0.096 µM to 0.98 µM) on lung cancer (A549) cell lines, and four compounds showed a broad spectrum of cytotoxic activity on all the examined cancer cells in the range of 0.096–4.58 µM. Among them, compounds **71** and **72** (Figure 27) were found to be the most active, with IC_50_ values range on all lines (except PC-3 for **71**) of 0.096–0.32 µM and with low cytotoxicity in the HeK-293T line, with IC_50_ values of 6.76 and 6.65 µM respectively.

In-depth studies of the impact of **71** and **72** on A549 lung cancer cells revealed a remarkable inhibition of cell migration through the disruption of the F-actin assembly. Moreover, treatment with **71** and **72** led to the collapse of the mitochondrial membrane potential (DJm) and increased the levels of ROS in A549 cells. Reported in [69] were the results of different studies suggesting that such hybrids have the potential to be developed as cytotoxic agents and their structural modifications could lead to a new generation of promising anticancer agents.

### 4.18. Imidazopyridine–4-Thiazolidinone Hybrids

Iqbal et al. [70] reported the synthesis of two series of imidazopyridine–4-thiazolidinone hybrid molecules and an evaluation of their anticancer activity on a panel of three human cancer cell lines: MCF-7 (human breast cancer), A549 (human lung cancer), and DU145 (human prostate cancer).

Among the synthesized hybrids, derivatives **73–75** (Figure 28) were found to be the most active, with IC_50_ values at the micromolar level. The preliminary SAR provided by the authors revealed that anticancer activity was connected to the R1 nature (Figure 28). The docking results suggested that the synthesized hybrids **73–75** could be potential EGFR kinase inhibitors.

## 5. Empirical SAR, Mechanisms of Action, and Molecular Targets for 4-Thiazolidinone-Bearing Hybrid Molecules with Anticancer Activity

Summarizing the structure–activity relationship analysis, the following trends can be observed. As reported in this review, 4-thiazolidinone-bearing hybrid molecules with anticancer activity mostly belong to the linked type of hybrids (Figure 29), and only molecules, 44 and 45, could be characterized as merged/fused hybrids. Hybrids mostly contain two or three potential pharmacophores (approved drugs and natural or privileged heterocyclic scaffolds) in their molecules. Most parts of the hybrids contain linked pharmacophores in positions C2 or C5, whereas substitution at the position N3 was used less often for design.

Applying imino/amino- and hydrazone-linkers between pharmacophores and a 4-thiazolidinone core at position two was often used in the design of novel hybrid molecules with potential anticancer activity (Figure 29). Good results were obtained using a pyrazoline linker at position two in the 4-thiazolidinone ring. It is worth noting that using of pyrazolines is a promising direction in the construction of hybrid molecules with anticancer properties. Pyrazolines can be considered cyclic bioisosteres of the hydrazone-linker which possess their own pharmacological profile and could be additionally substituted with potential anticancer pharmacophores. This makes pyrazoline scaffolds a desirable aspect of the hybrid molecules. In some cases, pharmacologically attractive substituents were linked to the C2 position of the 4-thiazolidinone core without a linker through the carbon–carbon bond. At the N3 position, as methylene-and amino groups were used as linkers, urea moiety or the linker was absent. Often position N3 was unsubstituted in the structures of reported molecules or was substituted with alkyl/aryl-groups or amino acids. The introduction of the enamine linker at C5 of the 4-thiazolidinone ring was used in molecules 43–45; however, in the case of hybrids 44 and 45, such a structure modification led to the disappearance of activity. The design using a ylidene linker at C5 was much more popular and resulted in the majority of reported hit/lead compounds.

The analysis of the mechanisms of anticancer effects reported in this review for hybrid molecules showed that they possess a complex mechanism of action and additional in-depth studies are needed in this area (Figure 30). 

A large number of hybrid molecules were characterized as potent apoptosis inducers via multiple mechanisms. Caspase-dependent (intrinsic) and extrinsic pathways, with the upregulation of anti-apoptotic signals and downregulation of pro-apoptotic signals, were the most often indicated. An important reported mechanism for antimitotic activity was an impact on the cell cycle with arrest following in different phases. Moreover, enzymes such as PIM-and tyrosine kinases as well as carbonic anhydrases were used as molecular targets for the design of novel 4-thiazolidinones with anticancer activity. Some reported hybrids were found to be promising tubuline inhibitors and inducers of ROS generation. 

## 6. Conclusions

The molecular hybridization strategy is a popular trend and an attractive research direction, providing endless source of significant opportunities in the design of new 4-thiazolidinone-bearing hybrid molecules with potential anticancer activity. A combination of potential pharmacophores in one molecule is one of the possible approaches used to achieve multi-target and polypharmacological effects for small molecules and is especially essential in the case of antimitotic agents. The hybridization of scaffolds, the hybrid-pharmacophore approach, and analogue-based drug design are key tools that are widely used by synthetic and medicinal chemists in this field. The hybridization of the 4-thiazolidinone core with early approved drugs, natural compounds, and privileged heterocyclic scaffolds is the most frequently used method in the development of novel molecules with antimitotic effects. The application of a molecular hybridization methodology allowed for identification of the new “hit” and “lead” hybrid compounds which target valid and important targets in carcinogenesis. The mentioned strategies and methodologies will remain relevant to studies over the next decade, and new wave/direction in the design of potential antimitotic agents using the hybridization of 4-thiazolidione scaffolds with biomolecules such as monoclonal antibodies, etc., can be expected. The data presented in this review contribute to the SAR profile of 4-thiazolidinone-based hybrid molecules for further exploration in the design, improvement, and optimization of new molecules with antitumor activity.

## Figures and Tables

**Figure 1 ijms-23-13135-f001:**
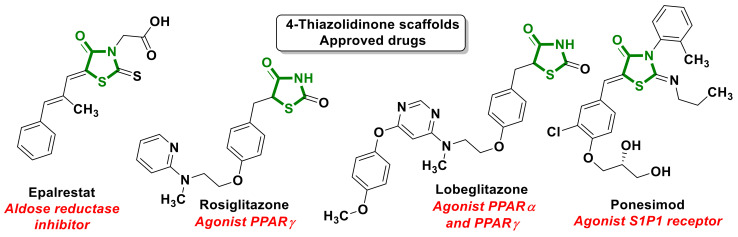
The profile of 4-thiazolidinone-bearing molecules in medicinal chemistry.

**Figure 2 ijms-23-13135-f002:**
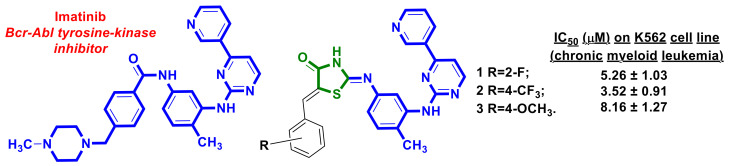
Structures of imatinib and 4-thiazolidinone-bearing imatinib analogs with anticancer activity.

**Figure 3 ijms-23-13135-f003:**
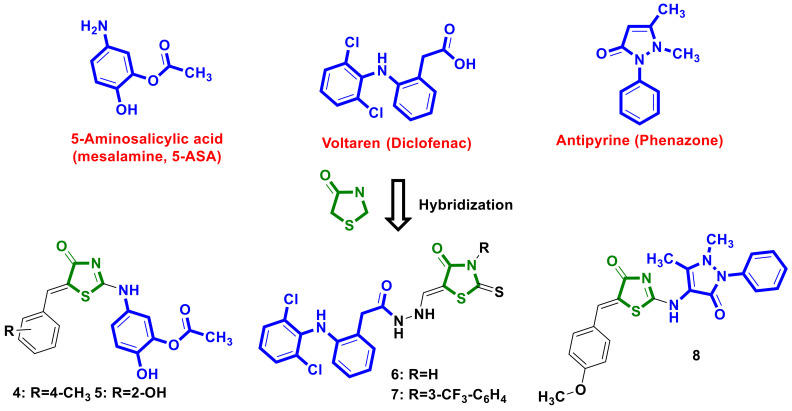
4-Thiazolidinone-bearing hybrids molecules with NSAIDs scaffolds possessing anticancer activity.

**Figure 4 ijms-23-13135-f004:**
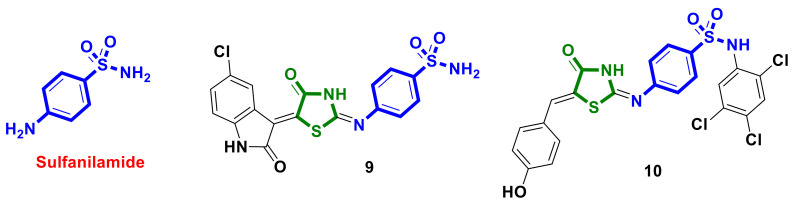
Sulfanilamide-bearing hybrids with 4-thiazolidinone scaffolds as potential anticancer agents.

**Figure 5 ijms-23-13135-f005:**
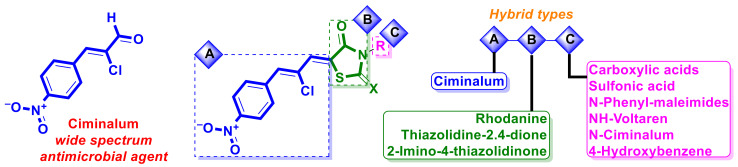
Structure of ciminalum and scheme of molecular hybridization strategy applied for design of potential anticancer hybrids with ciminalum moiety.

**Figure 6 ijms-23-13135-f006:**
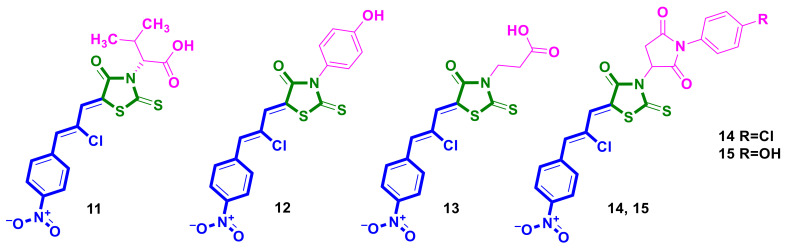
Ciminalum–4-thiazolidinone hybrid molecules as potential anticancer agents.

**Figure 7 ijms-23-13135-f007:**
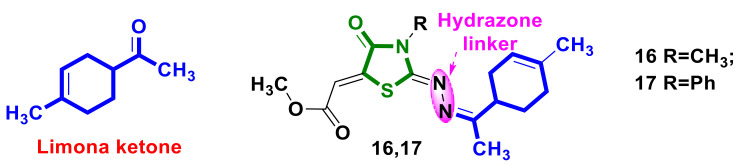
Structures of 4-thiazolidinone hybrids with limona ketone.

**Figure 8 ijms-23-13135-f008:**
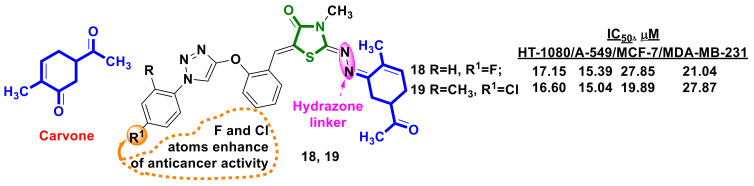
Structures of 4-thiazolidinone hybrids with carvone.

**Figure 9 ijms-23-13135-f009:**
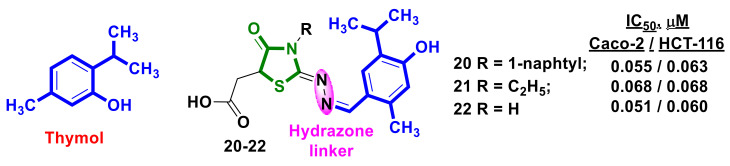
Structures of 4-thiazolidinone hybrids with thymol moiety in the molecules.

**Figure 10 ijms-23-13135-f010:**
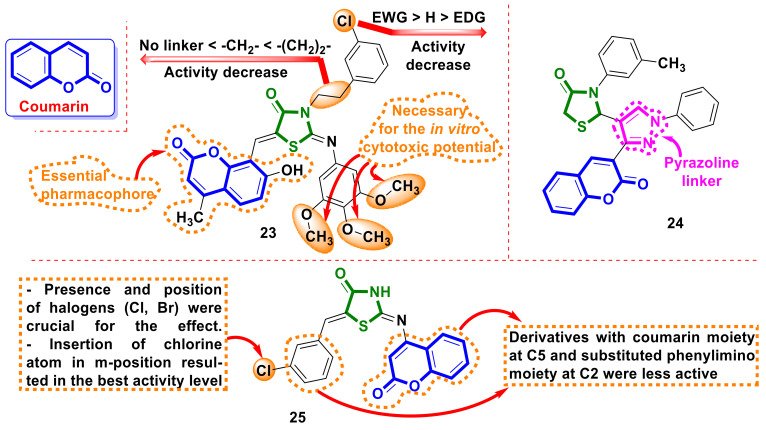
Structures of 4-thiazolidinone hybrids with coumarin motifs in the molecules and empirical SAR correlations.

**Figure 11 ijms-23-13135-f011:**
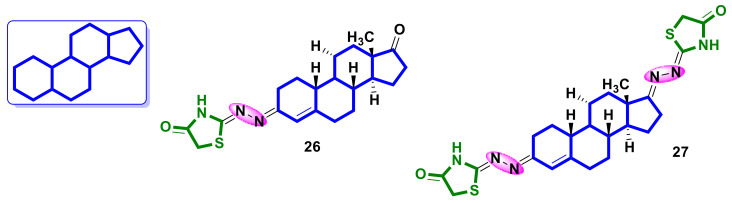
Structures of steroid–4-thiazolidinone hybrids.

**Figure 12 ijms-23-13135-f012:**
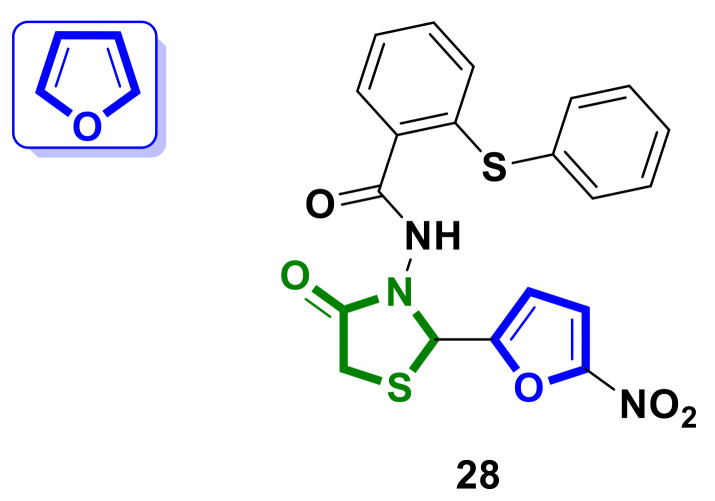
Structure of furan-bearing 4-thiazolidinone hybrid with potent anticancer activity.

**Figure 13 ijms-23-13135-f013:**
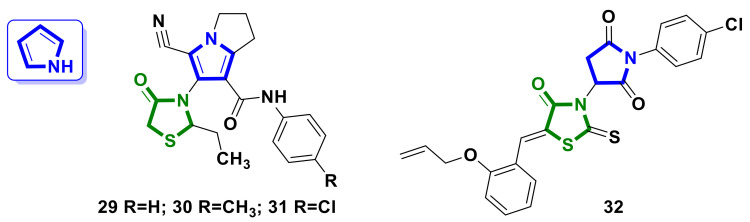
Structures of pyrrole-bearing hybrid molecules with 4-thiazolidinone scaffolds as potential anticancer agents.

**Figure 14 ijms-23-13135-f014:**
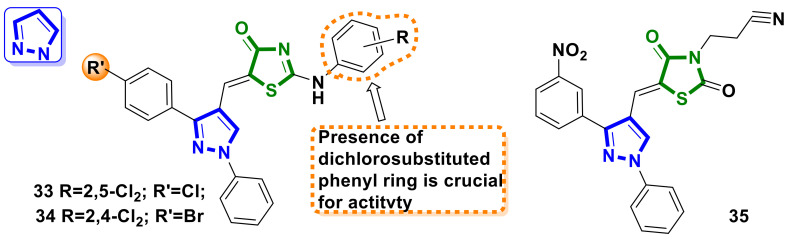
Structure of pyrazole–4-thiazolidinone hybrid molecules as potential anticancer agents.

**Figure 15 ijms-23-13135-f015:**
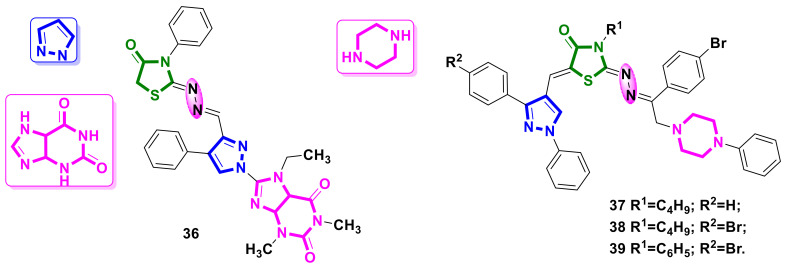
Pyrazole-purine- and pyrazole-piperazine–4-thiazolidinone hybrids as potential anticancer agents.

**Figure 16 ijms-23-13135-f016:**
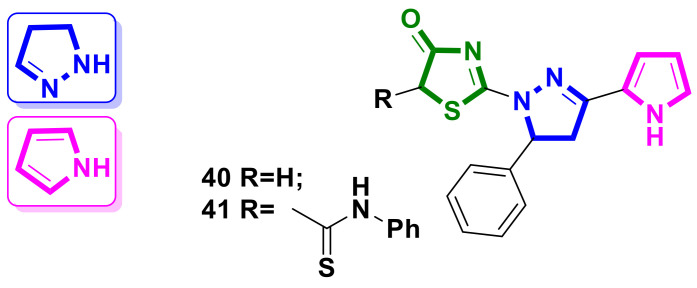
Structure of pyrazoline-pyrrole–4-thiazolidinone hybrid molecules.

**Figure 17 ijms-23-13135-f017:**
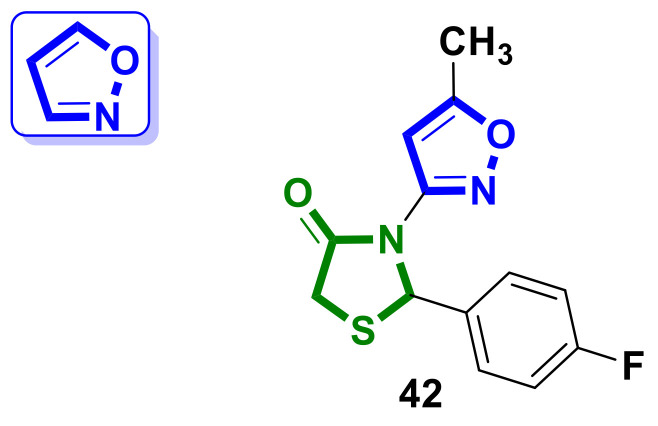
Structure of isoxazole–4-thiazolidinone hybrid molecule.

**Figure 18 ijms-23-13135-f018:**
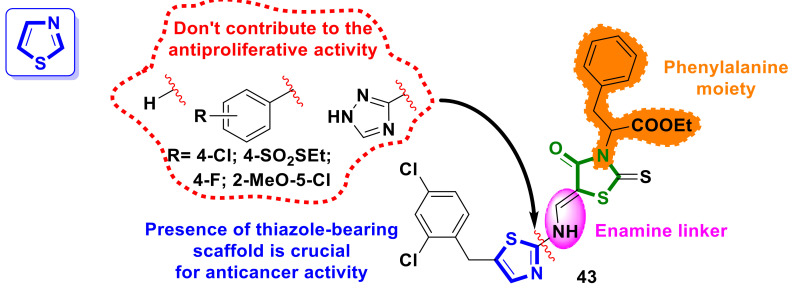
Structures of thiazole–4-thiazolidinone hybrids with antimitotic activity and SAR correlations.

**Figure 19 ijms-23-13135-f019:**
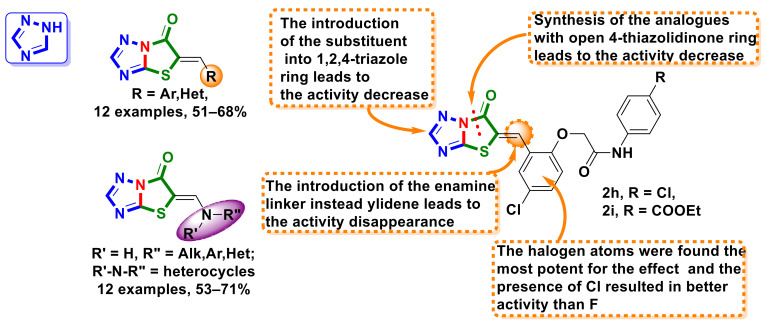
Structures of merged 1,2,4-triazole–4-thiazolidinone hybrids with antimitotic activity.

**Figure 20 ijms-23-13135-f020:**
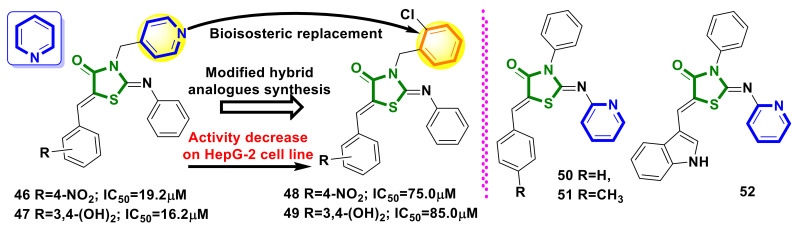
The structures and SAR of pyridine–4-thiazolidinone hybrids with anticancer properties.

**Figure 21 ijms-23-13135-f021:**
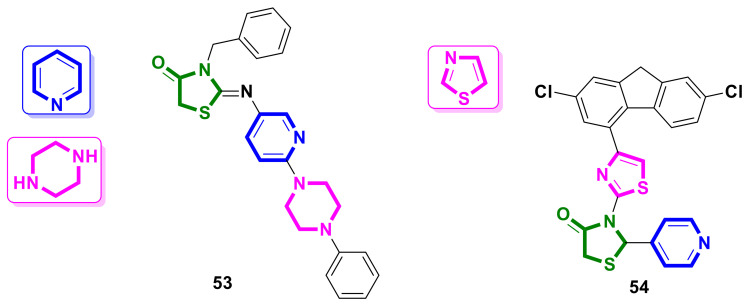
The structure of hybrids containing pyridine, piperazine, thiazole, and 4-thiazolidinone scaffolds in the molecules.

**Figure 22 ijms-23-13135-f022:**
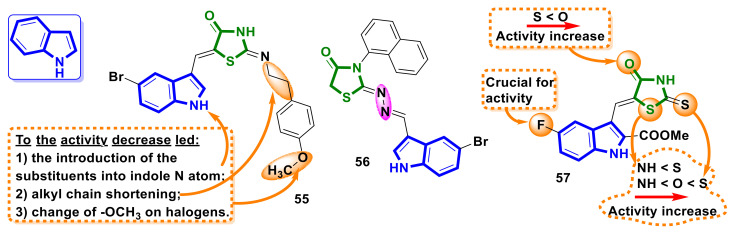
The structures of indole-bearing hybrids molecules with 4-thiazolidinone scaffolds as antimitotic agents.

**Figure 23 ijms-23-13135-f023:**
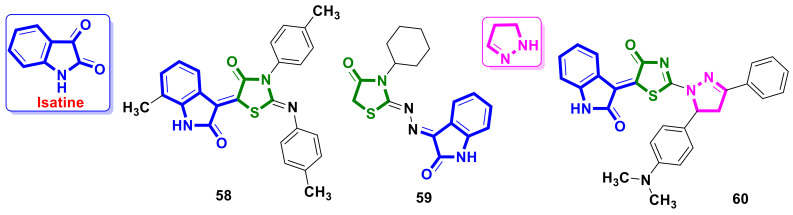
Isatine-based 4-thiazolidinone hybrid molecules with anticancer activity.

**Figure 24 ijms-23-13135-f024:**
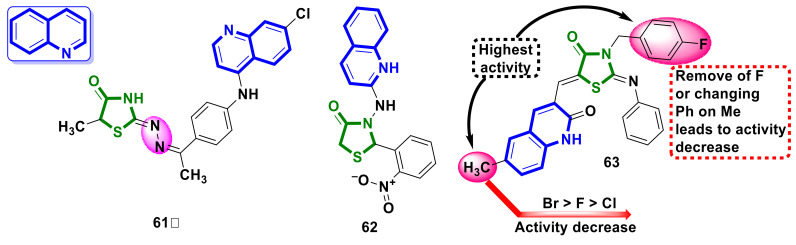
Quinoline-bearing hybrid molecules with 4-thiazolidinone scaffolds as potential anticancer agents and selected SAR correlations.

**Figure 25 ijms-23-13135-f025:**
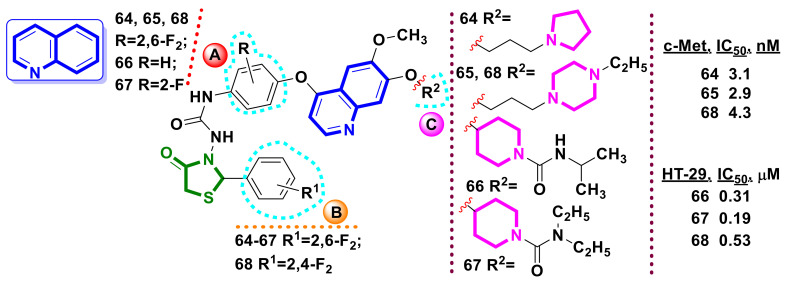
Quinoline-bearing hybrid molecules with 4-thiazolidinone scaffolds as potential anticancer agents.

**Figure 26 ijms-23-13135-f026:**
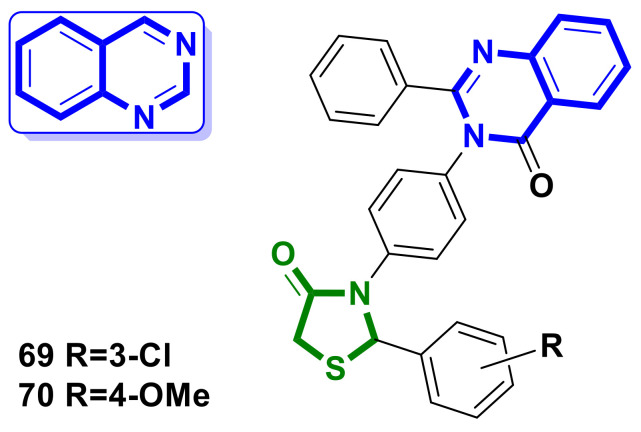
Quinazoline-bearing hybrid molecules with 4-thiazolidinone scaffolds as potential anticancer agents.

**Figure 27 ijms-23-13135-f027:**
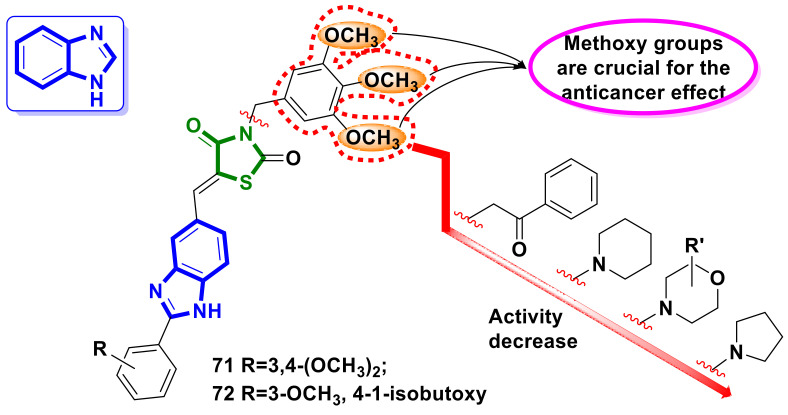
Benzimidazole–4-thiazolidinone hybrid molecules as potential anticancer agents and key SAR points.

**Figure 28 ijms-23-13135-f028:**
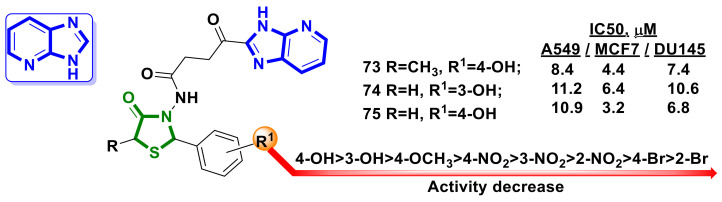
Imidazopyridine–4-thiazolidinone hybrid molecules as potential anticancer agents.

**Figure 29 ijms-23-13135-f029:**
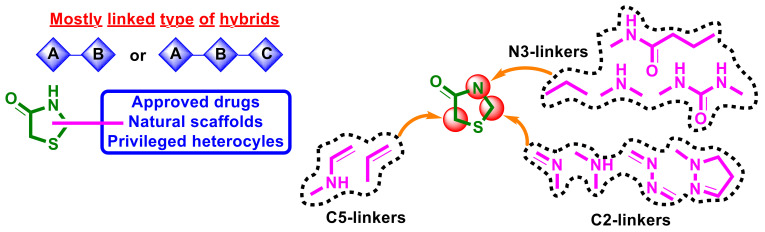
Schematic presentation of empirical SAR for 4-thiazolidinone-bearing hybrid molecules with anticancer activity.

**Figure 30 ijms-23-13135-f030:**
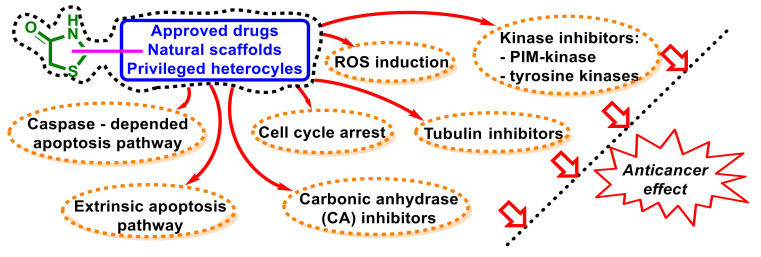
Schematic presentation of mechanisms of action and molecular targets for 4-thiazolidinone-bearing hybrid molecules with anticancer activity.

## Data Availability

Not applicable.

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
