# Peer review of "4-Thiazolidinone-Bearing Hybrid Molecules in Anticancer Drug Design"

_ijms, 2022, doi:10.3390/ijms232113135_

Round 1

Reviewer 1 Report

This review manuscript by Roszczenko et al. describes “4-Thiazolidinone-bearing Hybrid Molecules in Anticancer Drug Design.” I would recommend this manuscript to be published after major comments as below

1.   Rewrite first sentence of the abstract. 

2.    What was the search strategy used to search articles to write this review? This can be included in the introduction section.

3. As this review summarizes articles from 2017-2022, please summarize these articles in this review, Chemico-Biological Interactions 262 (2017) 46-56; Journal of the Iranian Chemical Society , 19, 793–808 (2022); Arch Pharm. 2020;353:e2000071; Polycyclic Aromatic Compounds, DOI: 10.1080/10406638.2022.2108074; European Journal of Medicinal Chemistry 138 (2017) 234-245; Bioorganic Chemistry 86 (2019) 126–136; J Biomol Struct Dyn 2021 Dec 20;1-12; J. Heterocyclic Chem.,56, 1794 (2019)

4.  Wherever IC50 is not included in the text, please write those values either in the text or below each structure. Page 3, for reference 16 and 21, at least percentage inhibition should be added in the text. Same should be followed for all summaries wherever it’s not written.

5. Authors should write about structure-activity relationship in summaries and some of the key important points observed in SAR.

6. On page 10, line 325, it should be a difference reference than 31 and I think this article should be Future Med. Chem. (2017) 9(15), 1709–1729.

7. Authors should write about future perspectives of thiazolidinone-based scaffolds in the conclusion section.

Author Response

Response to the Reviewer 1:

According to the Reviewer’s suggestions:

  1.  Rewrite first sentence of the abstract. 

The sentence has been changed, as suggested.

  1. What was the search strategy used to search articles to write this review? This can be included in the introduction section.

In the introduction, we included an article search strategy that was based on searching databases such as Scopus (Elsevier), SciFinder (Chemical Abstracts), and PubMed.

  1. As this review summarizes articles from 2017-2022, please summarize these articles in this review, Chemico-Biological Interactions 262 (2017) 46-56; Journal of the Iranian Chemical Society , 19, 793–808 (2022); Arch Pharm. 2020;353:e2000071; Polycyclic Aromatic Compounds, DOI: 10.1080/10406638.2022.2108074; European Journal of Medicinal Chemistry 138 (2017) 234-245; Bioorganic Chemistry 86 (2019) 126–136; J Biomol Struct Dyn 2021 Dec 20;1-12; J. Heterocyclic Chem.,56, 1794 (2019)

We have added the reviewer's suggested articles, thank you for pointing out these gaps, as it has enriched our article.

  1. Wherever IC50is not included in the text, please write those values either in the text or below each structure. Page 3, for reference 16 and 21, at least percentage inhibition should be added in the text. Same should be followed for all summaries wherever it’s not written.

We added the missing values.

  1. Authors should write about structure-activity relationship in summaries and some of the key important points observed in SAR.

Missing data were completed, in addition, a chapter summarizing the issue was written.

  1. On page 10, line 325, it should be a difference reference than 31 and I think this article should be Future Med. Chem. (2017) 9(15), 1709–1729.

Thank you for bringing it to our attention, the reference has been corrected.

  1. Authors should write about future perspectives of thiazolidinone-based scaffolds in the conclusion section.

We have included information on future perspectives in the conclusions.

We would like to thank the Reviewer for the valuable comments and suggestions. Accordingly, we have revised and tried our best to improve the manuscript. We sincerely hope that the revised manuscript will meet your approval.

Reviewer 2 Report

Roszczenko and co-workers summarize 4-thiazolidinone-bearing hybrid molecules in anticancer drug discovery. The idea to combine active motifs is good but poorly executed. The manuscript is poorly organized and full of linguistic errors. Several sentences are broken and misleading. Authors should elaborate the key information and draw an SAR pattern. Correct structure in Figure 12 (nitro group).   

Author Response

Response to the Reviewer 2:

The authors would like to thank the Reviewer for taking the time to review our paper. According to the Reviewer’s suggestions:

  1. Roszczenko and co-workers summarize 4-thiazolidinone-bearing hybrid molecules in anticancer drug discovery. The idea to combine active motifs is good but poorly executed. The manuscript is poorly organized and full of linguistic errors. Several sentences are broken and misleading.

    Thank you for pointing out the language errors, we have corrected the ones we were able to find. In addition, we have organized the manuscript, so we hope that after the corrections it will meet with approval.

  2. Authors should elaborate the key information and draw an SAR pattern.

A separate chapter has been created for SAR analysis.

  1. Correct structure in Figure 12 (nitro group). 

The figure has been corrected. 

The authors would like to thank the Reviewer for the thorough analysis of the paper and the valuable comments that significantly increased the scientific value of the article. We hope that the revised paper will meet your approval.

Round 2

Reviewer 1 Report

Authors revised manuscript by including IC50 values, adding structure-activity relationships, summarizing articles as per suggestions. I would recommend this manuscript to be published after minor additions and correction as below

1.  For reference 33 and 35, SAR could be added.

2. Reference 69, these compounds are apoptotic agents not antimitotic agents (as authors did not test antimitotic activity in this paper).

Author Response

Response to the Reviewer 1:

According to the Reviewer’s suggestions:

  1. For reference 33 and 35, SAR could be added.

We have added a figure containing the SAR, as recommended by the reviewer.

  1. Reference 69, these compounds are apoptotic agents not antimitotic agents (as authors did not test antimitotic activity in this paper).

We have made the change, thank you for pointing out this mistake.

We would like to thank the Reviewer for the valuable comments and suggestions. Accordingly, we have revised and tried our best to improve the manuscript. We sincerely hope that the revised manuscript will meet your approval.

Reviewer 2 Report

Authors have significantly improved the manuscript. It could be considered now.

Author Response

Response to the Reviewer 2:

The authors would like to thank the Reviewer once again for the valuable comments that significantly increased the scientific value of the article.
